# Drugs Becoming Generics—The Impact of Genericization on the Market Performance of Antihypertensive Active Pharmaceutical Ingredients

**DOI:** 10.3390/ijerph18189429

**Published:** 2021-09-07

**Authors:** Bence Kovács, Miklós Darida, Judit Simon

**Affiliations:** 1Institute of Marketing, Corvinus University of Budapest, 1093 Budapest, Hungary; judit.simon@uni-corvinus.hu; 2Global Pipeline and Portfolio Project Management Department, Gedeon Richter Plc., 1103 Budapest, Hungary; 3Medical Division, Gedeon Richter Plc.,1103 Budapest, Hungary; daridam@richter.hu

**Keywords:** pharmaceutical therapies, market performance, genericization, diffusion, Europe

## Abstract

To explore long-term changes in intra and inter-class choices between generic compounds, this paper investigates the market trends of two antihypertensive drug classes that have closely related pharmacological mechanisms—angiotensin convertase enzyme inhibitors (ACEIs) and angiotensin receptor blockers (ARBs). We analysed the development of ACEI and ARB markets between 2001 and 2016 in nine European countries, covering the genericization transition periods of both therapeutic groups. The analysis was undertaken on the level of the active pharmaceutical ingredients (API) and focused on international and country-specific diffusion patterns. Comparison of ARB and ACEI therapies shows that although ARBs became off-patent during the observed period, and have a clinical advantage in terms of the adverse event profile over ACEIs, the increasing dominance of ARBs cannot be identified. One explanation is that ACEI therapies became generics earlier, relocating competition to the level of brands, while competition among ARBs remained at the level of the APIs. As for intra-class drug preferences, it was observed that the long-term trends show that ramipril outperformed its ACEI competitors, even though the kinetics and the rank order of preferred active compounds were inconsistent among markets. The diffusion of clinically preferable therapies seems to be ultimately supported by generic entries. In Eastern European countries, the emergence of generic markets has not only improved access to ACE inhibitors and ARBs, but has been a prerequisite for changing preferences. In contrast, genericization resulted in the relative anchoring of prior, branded era-based preferences in some Western European countries, which may be attributed to the role of the cessation of promotion and the fixity of prescription behaviour.

## 1. Introduction

### 1.1. Introduction

Ideally, when choosing a therapy, physicians select the option that ensures the best outcome for patients. By the time a whole class of drugs becomes off-patent, significant scientific knowledge and medical experience has been gathered about each individual active pharmaceutical ingredient in the group, which theoretically permits an informed, evidence-based prescription choice, even if treatment guidelines formulate class-level recommendations. While the impact of patent losses on pharmaceutical spending and switching from branded products has been of intense scientific interest, less is known about how the generic markets of specific drug classes develop in the long term. In relation to the two drug groups under analysis, the first-in-class ACE inhibitor captopril was approved in 1981, followed by a dozen other active moieties, while in terms of ARBs, losartan was approved in 1995, followed by several active ingredients. This abundance of active ingredients provides an appropriate basis for our research.

In our analysis, we accept the principles of the evidence-based medicine (EBM) approach and investigate whether diffusion is more likely to happen with therapies with more favourable clinical profiles [1,2]. While the EBM model represents an appealing framework that combines global scientific knowledge with individual insights and needs, some practical limitations should be kept in mind. New medical advances and study findings become available every day; however, the continuous incorporation of fresh medical information is hindered by various phenomena in clinical practice [3,4].

In the article from 2017, Kovács and Simon analysed the effect of clinical evidence and price on prescription drug sales, hypothesizing that generic markets favour those drug classes or active moieties from direct competitors that represent the highest therapeutic value. Their assumption, based on EBM principles, was that an absolute clinical ranking (Appendix A, Figure A1) can be defined based on the clinical differences among the active ingredients used in the same indication [5]. The findings of the article assume a positive correlation between clinical evidence and market performance. However, the authors drew their conclusions based on data about ACEI and ARB sales in a single year.

Realistically, the sales or the market position of a medicine, drug class, or pharmaceutical substance cannot be expected to remain unchanged over time. In the present study, we explore and describe the dynamics of the selected generic markets, investigating whether preferences for ACE inhibitors and ARBs have changed over time on a macro level. Besides enquiring about the changes in the overall use of active moieties and drug classes, the authors intend to obtain insights about prescribing characteristics, with special attention paid to the following questions:○In terms of long-term trends, how does clinical evidence affect market performance?○Have preferences for groups of drugs and for active ingredients been affected by the genericization of ARBs and ACE inhibitors in European markets?○Can international or country-specific interrelationships be identified based on long-term prescribing trends?

### 1.2. The Life Cycle of Drugs and the Market Characteristics of Generic Drugs

The pharmaceutical market is driven by constant innovation, yet the pharmaceutical market cannot be simplified to the interaction between the supply of innovative drug manufacturers and the demands of patients. As for the manufacturing side, in addition to innovative companies, competition among generic players is becoming increasingly intense. While generic drugs are gaining ground, drug policies that aim to cut costs are becoming widespread. Although a significant proportion of drugs that are designed to treat common diseases such as hypertension and diabetes have lost their patent protection, the range of available therapies has also changed in recent years.

Drug development and drug pricing are also defined by law and regulations [6,7]. There are many regulatory techniques for affecting both demand and supply, in relation to which national drug policy makers expect that costs are controlled, funding sustained, prices decreased, and access to medicines improved [8].

In addition to those drugs that are new from every perspective (i.e., first-in-class) [9], other original drugs with different active ingredients but a similar mode of action can also be introduced to the market.

In principle, me-too and follow-on drugs increase the therapeutic potential of the market, contributing to well-being and price reductions; furthermore, their approval encourages further market development. However, some believe that they fail to represent genuine innovation, and that price competition may not be effective, as in certain cases, they may lead to greater expenditure [10,11,12].

As long as patents provide protection (typically 20 to 25 years), or data exclusivity or market exclusivity is maintained, generic competitors cannot be introduced to market. As innovation costs are huge, the time it takes for an innovator to launch a generic drug to the market is enormously important, and during this period, their investment can be recouped [13,14].

With the end of the monopoly on an original product, generic drugs basically act to lessen spending on pharmaceuticals. Price reductions can lead to better access to drugs and an increase in sales of an active ingredient [15,16]. However, it should not be generalized that generic drugs inevitably mean price reductions, or that with the start of generic competition, the perfect principles of free competition prevail. With the entry of generic drugs, the price of an original branded product typically remains the same or even increases (the “paradox of generics”). This may be attributed to prescribers susceptible to pharmaceutical promotion continuing to prescribe the original brand [17].

The speed of any decrease in price after the loss of exclusivity depends on the price prior to the expiry of patents and the size of the market, both of which factors define how attractive generic companies find it to enter the market with a cheaper version of a product [18]. In very small markets, it may not be attractive for a generic manufacturer to launch a generic product, thus the innovator can remain the sole distributor of a drug for a long time [19]. Thus, the average duration until the entry of the first generic drug varies from country to country, just as the proportion of prescriptions of generics does (“generic penetration”) [20]. Furthermore, with the acquisition of competitors, generic companies may reduce competition. In extreme situations, this may result in a monopoly situation and drastic price increases [21].

To protect their markets, innovative manufacturers attempt to defer the market entry of generic drugs in different ways. In addition to obtaining additional market protection (e.g., new registrations, or the registration of paediatric indications), the past few decades have seen various attempts at improving competitive positions. An example of this is the introduction of a new, patent-protected drug by family extension, and switching patients before genericization. In this way, a part of an innovator’s market can thus be protected (“product hopping/evergreening”) [22,23].

Furthermore, several such methods have been used that raise concerns related to different areas of competition law. A good example is patent manipulation—when an innovator attempts to make market entry difficult by using patents that fail to involve significant innovation, or the innovator tries to reach a patent settlement with manufacturers, aiming to enter the market to prevent the latter from attacking their patents and bringing their own products to market [22,23]. In the European market for ACE inhibitors, such practices—generic competition undermined by patents—have been identified. Servier, the developer of perindopril, has filed numerous patent applications that lack genuine innovation aimed at preventing the market entry of generic drugs, also seeking to obtain circumventing technologies. The company attempted to reach patent agreements with pharmaceutical companies considering entry to the generic market between 2005 and 2007. Finally, the European Commission ordered Servier and the companies with which Servier had reached patent settlements to pay a fine of over EUR 400 million for violating competition law [24].

As cheaper bio-equivalent drugs may reduce costs and improve access to therapies, health policy strongly prioritizes the promotion of generic drugs. Therefore, measures for promoting the use of generic drugs can often be identified in relation to regulations applied to pharmaceutical markets. These include, among other areas, encouraging generic development (e.g., public authorities issuing development guidelines), substitution by generics in the distribution and prescription process, ordering drugs according to their international non-proprietary name (INN), or incorporating financial incentives into the financing of physicians’ and pharmacists’ practices. Methods used to subsidize medicines and drug pricing vary from country to country, but a very frequently used and somewhat controversial method is so-called reference pricing. This involves the price of a subsidized preparation being tied to its cost in another country (“external reference pricing”) or to the price of similar drugs (“internal reference pricing”) [8,25]. However, following the economic crisis of the 2000s, an increasing number of countries considered it important to cut back on drug spending, and thus health policy measures aimed at promoting the prescription of generic drugs are increasingly being used [26].

### 1.3. Factors Influencing the Choice of Drug Therapies and Drug Sales

Prescription drug sales are basically defined by the following factors: the number of prescriptions filled out by physicians; whether the prescriptions are actually dispensed; and what kind of products patients receive for their prescription in pharmacies. On the basis of the latter, Danzon and Furukawa proposed distinguishing between two types of generic market: physician-defined markets, where physicians can decide which product from which generic manufacturer to prescribe; and pharmacy-driven markets, where the pharmacy basically decides which active ingredient from which manufacturer to sell [27]. In our analysis, the UK can be located in the latter category, while in Italy, Germany, Spain and France, physicians’ prescribing decisions have been found to be the main factor. Partly due to the drug policy measures that were the result of the economic crisis of the late 2000s, the distinction between the two approaches started to blur in several countries. At the very least, the latter changes strengthened the role of pharmacies [28]. In 10 EU countries, mandatory active-ingredient-based prescription or mandatory generic substitution had been introduced into the legal system by 2016. In other EU countries, this was encouraged in different ways [29].

There are many publications about the mechanisms involved in and motivations for medical prescription-related decisions and the market introduction of new therapies. In these publications, various factors that may potentially influence the diffusion of new therapeutic options are identified at the micro- and meso-levels. These include physician-related factors (socio-demographic status, scientific profile, prescription habits, exposure to various forms of promotional activities and contagion through social networks), characteristics of the medical practice, types of patients treated, and medication. The drug-related factors affecting such decisions may be directly measurable ones (such as the marketing expenses of the manufacturer, general acceptance of the drug, therapeutic novelty, number of competitors, or price of the drug) in addition to actual medical characteristics (addressing unmet needs, advantage over available alternatives, or safety-and-efficacy-related perceptions) [30,31,32].

Rogers’ diffusion model shows that innovation diffuses across societies through various communication channels over time [33]. Plotting the proportion of new technology users over time, the model describes (by default) a sigmoid curve, as it distinguishes five groups—innovators, early adopters, early majority, late majority and laggards—based on their attitude towards a new process (in our case, towards a new active ingredient) [34]. At the individual level, five stages of the adoption process are distinguished—from learning about the new opportunity to committing to its use: knowledge/awareness, persuasion, the decision to reject or adopt, implementation, and confirmation. Following this approach, Rogers’ model combines macro-level processes with micro-level events, which, when applied to the pharmaceutical market, can lead from individual medical therapeutic decisions to the market penetration of innovative drugs.

With regard to specific therapeutic options and the choice of active ingredients, Denig attributed a decisive role to the amount of medication recommended by physicians (the evoked set). According to her model, physicians either attempt to select prescribed medication out of habit, or actively seek the solution to a clinical question. The impact of the decision will influence the decision-making process and will be incorporated into experience and knowledge—the basis of future therapeutic choice—and drug selection habits. According to the studies Denig conducted among Dutch hospital doctors in the 1990s, the size of the evoked set was between 1.7 (platelet aggregation inhibitors) and 5 (antihypertensive drugs) on average, depending on the group of drugs [35].

An article about research into the marketing strategy of ACE inhibitors stated as early as in the early 1990s that Servier’s perindopril (Coversyl) also affected national preferences among French prescribers [36]. The manufacturers of the 14 ACE inhibitors present on the branded market attempted to distinguish themselves from other members of the group with various messages: with regard to older drugs (captopril, enalapril), by primarily emphasizing new indications (heart failure and heart attack post condition, and diabetes-related hypertension); by referring to new entrants (e.g., ramipril, perindopril, quinapril), by stressing their success at preventing organ damage; while very recent ones (trandolapril, benazepril) have attempted to build their communication more on aspects of convenience.

Although drugs that act on the same biological target show strong similarity in principle, due to their different chemical structures, they may behave differently in the human body, whether pharmaco-kinetically or pharmacodynamically. Besides this, it may not be possible to obtain the same clinical evidence for each active ingredient. This is the case with ACE inhibitors, too. Although all these drugs were registered for the treatment of hypertension, not all drugs are associated with clinically based supporting data about their effects on other conditions, such as improving the survival chances of heart failure or heart attack [37,38,39]. From a practical point of view, there may be supporting arguments for formulating group-level recommendations, because guideline messages can be simplified and clarified, and the amount of data that practitioners need to remember can be reduced. Nonetheless, in their investigation into prescription data of OECD countries, Maggioni et al. revealed that nearly a quarter of heart failure patients either did not receive ACEI or ARB treatment, or not in the right dose, in opposition to the suggestions of the treatment guidelines [40].

Concerns have been raised about the quality of information provided by the pharmaceutical industry in their marketing [41,42,43], and there is evidence that physicians tend to be critical about the pharmaceutical industry either in the scientific literature [44] or even among their colleagues [45]. Nonetheless, promotional activity is certainly a factor that should be considered in drug prescription decisions and in the processes of the pharmaceutical market, even if the effects of such promotion are somewhat contradictory. A number of studies indicate that marketing tends to have a detrimental effect on the quality of prescriptions but increases the demand for medicines, while other studies found no significant correlation between these factors. The only conclusion that Spurling et al. were able to come to after reviewing the relevant literature was that there is no evidence that prescription habits can be improved by promotion [46]. It is interesting from the perspective of this study that Greving et al. found in their research among Dutch physicians about antihypertensive drugs in the early 2000s that physicians who tend to rely more heavily on pharmaceutical industry information were more likely to start to prescribe ARBs to their patients [47].

According to Venkataraman and Stremersch, marketing effectiveness is modified by the efficacy and side effects of a drug [48]. The picture is further complicated by the fact that it does matter what kind of information is made available to physicians in the various life stages of the product, because emphasizing a certain product characteristic in a competitive environment can be beneficial or detrimental to product sales [49].

An interesting feature of drugs is that they are experience goods. Namely, whether individual drugs are good for patients transpires only after they have been taken. Crawford and Shum, in their study on anti-ulcer drug prescriptions, concluded that patients were fundamentally risk-averse and that initial uncertainty about drugs quickly disappeared, so they were not interested in switching drugs and continued treatment with the first drug of their choice. These authors also called attention to the fact that marketing that creates highly positive consumer perceptions may lead to market concentration, even if the alternatives that are available are essentially similar [50].

## 2. Materials and Methods

### 2.1. Data Used in the Investigations

Information concerning market performance was obtained from the IQVIA Health MIDAS database made available by Gedeon Richter Plc. IQVIA MIDAS data combine country-level data, healthcare expertise and therapeutic knowledge in 90+ countries to deliver data in globally standardized forms to facilitate multi-country analyses, acting as a leading source of insight into international market dynamics relating to the distribution and use of medicines. IQVIA MIDAS data are designed to support multi-country analyses of trends, patterns and similar types of analyses. All of the calculations, algorithms and methodologies used to produce these estimates of real-world activity makes the data highly reliable for these intended uses.

Quarterly aggregate data were obtained for the period from 2001 to 2016 on ACE inhibitor and ARB sales broken down into two-to-five-year periods for nine countries (France, Germany, Hungary, Italy, Poland, Romania, and the Netherlands; sales figures for the latter were available from 2004 onwards only). The selection of countries for the descriptive analysis was based on the cross-sectional study by Kovács and Simon [5]. The database contained drug class, active ingredient, and brand data, providing data about the sales of marketed branded and generic products from different manufacturers containing the same active ingredient. We retrieved data from the database for the main group of C09 ATC entirely according to the brand and mechanism of action, thus including within the two groups of mechanisms both drug products with a single active component and drug combination products (codes C09A and C09B, as well as codes C09C and C09D at ATC level 4). On the first level of query, distribution by form of dosage was examined. Since 99.9% of ACE inhibitors are taken orally worldwide, only oral pharmaceutical forms were filtered during later queries. Therapeutic group–country-, active pharmaceutical ingredient–country-, and active pharmaceutical ingredient–country–brand-level queries were also undertaken for the investigated markets. To indicate pharmaceutical sales, manufacturer revenues (thousand euros, MNF) and sales volumes (thousand ‘counting units’, CU; that is, pills, capsules, sachets, etc.) were taken into account in subsequent calculations. In order to ensure comparability, we analysed sales volume data in the relevant countries: more precisely, we compared the sales ratio that describes each API’s market performance in each country under investigation. We also took into account the fact that the APIs differ concerning defined daily dose (DDD), and that different strengths are available. Days of treatment (DOT) can be known only if we have all of the mentioned information about the APIs. We retrieved information about sales volume (thousand tablets) from the IMS database in the following format: API-strength-country. From these data, we were able to calculate whether the ratio of sales volume (later defined as CU/MAT) was linearly proportional to DOT. We confirm that the values used in the analysis represent the ratio of DOT of the API therapies. To correct for differences in posology among compounds, the daily defined dose (DDD, the assumed average maintenance dose per day for a drug used for its main indication in adults) was calculated on the basis of DDD data published by the WHO, and DDD correction was applied to sales data [51,52]. Sales data available for each brand were aggregated at the active ingredient level for each ATC group, and quarterly data were summarized annually.

To estimate prices, manufacturer revenues and volume (CU) data were used, and prices were estimated for DDD-adjusted quantities separately. This estimate does not provide consumer prices, but the aim was to analyse aggregate data at the active ingredient level and to examine the long-term relative market performance of each active ingredient in different countries. As the demand of the pharmaceutical market is the outcome of decisions taken by several players on the consumer side, it seemed to be more appropriate to estimate the manufacturer prices of active ingredients. Data were processed with Microsoft Excel 2010 and Stata IC 13.1 software (StataCorp LLC Lakeway Drive, College Station, TX, USA).

### 2.2. Multidimensional Scaling and Indicators for Interpretation

Subsequent to the necessary data conversion, multidimensional scaling was undertaken to analyse drug sales of the C09 ATC group, applying the methodology suggested by Kovács and Simon for both single-drug formulations and combination drugs. Multidimensional scaling is a method that facilitates the comparison of objects based on the level of similarity while taking multiple variables into consideration. The method is capable of exploring the structure of data in such a way that objects are represented as geometric relationships among points in a multidimensional space. The advantage of the method is the resulting graphic display that illustrates the magnitude of differences between the objects, showing which ones are close to each other. The statistical reliability and validity of the solution is measured by the value of R2 and a stress indicator. The method does not provide a direct solution for interpreting the dimensions of the space of perception and object characteristics. We interpreted the results based on professional experience and by collating the multidimensional scaling output with the indices derived from sales data. In our analysis, countries were located as objects in the multidimensional space, and similarity data were derived from their characteristics [53,54]. Aiming to compare the data of nine countries, the analysis was performed by the “classical” method available in Stata on annual DDD-adjusted data for the different years.

In data analysis, clinical and pharmacy sales data, separately recorded in the database, were summarized at the product group- and at the active ingredient-level; that is, both clinical and retail sales data were considered. We calculated ARB and ACEI market characteristics from raw data, and we worked out the input index numbers for this group as follows:ARB: ACEI price level ratio: ratio of ARB price level and ACE inhibitor price level, calculated as the quotient of manufacturer sales revenue and sales volume;ARB: ACEI volume ratio: ratio of ARB sales volume and ACE inhibitor volume;ARB: ACEI sales revenue ratio: manufacturer sales revenue of ARBs and ACE inhibitors;ARB preference index: ‘ARB: ACEI volume ratio’ (2) multiplied by ‘ARB: ACEI sales revenue ratio’ (3). We used this variable as a composite index to estimate the bias towards ARB use in relation to ACEIs despite the higher price levels.

The analysis was undertaken based on data for both single-drug formulations and fixed-dose combination drugs. This article presents the results based on data that represent all market, mono, and combination therapies taken together.

### 2.3. Approach to Analysing Changes in ACE Inhibitor and Arbs Sales Data over Time

To examine changes in market characteristics (namely, the relative sales data of individual countries over time), we created the indices described above for data associated with the years 2001, 2009, and 2016 at the beginning, middle, and end of the data series, respectively. Single-drug formulations and combination drugs were also taken into account. To visualize the changes, multidimensional scaling was undertaken.

Taking an explorative approach, we further investigated the market processes of the active ingredients of ACE inhibitors, focusing on three of the above-mentioned Central and Eastern European countries (Hungary, Poland, and Romania), in addition to three Western European countries (France, Germany, and the UK).

To illustrate market competition, prices and the number of brands of the same active ingredient concurrently present in the market were plotted, as changes in prices and changes in the number of available substitutes may indicate the start of generic competition. To characterize the market concentration of ACE inhibitors, we calculated the Herfindahl–Hirschman concentration index (both DDD and non-DDD adjusted HHI) for the selected years and plotted it versus time. The analysis of relative market shares of ACE inhibitors together with HHI calculations on non-DDD-adjusted data provided an overview of the relative sales of individual dosage units (tablets, capsules, etc.) regardless of the doses applied.

## 3. Results: Development of Preferences for ARBs and ACE Inhibitors in Nine European Countries

### 3.1. Market Features of ARBs and ACE Inhibitors in 2016, 2009 and 2001

In order assess the market performance trends of the two groups of antihypertensive therapies, a short summary of their history should be given. Of the two mechanisms of action, ACE inhibitors were the first therapies to be approved and launched from the beginning of the 1980s (captopril, enalapril in 1980–1981, lisinopril, perindopril and ramipril in 1987–1988–1989, respectively; other molecules followed afterwards). ACE inhibitors lost their patent exclusivity mainly in the 1990s and early 2000s, and manufacturers tried to prolong patent protection even through violations of the law (see the perindopril case referenced above). The first ARB’s market entry occurred in 1995 with losartan (valsartan in 1996, candesartan in 1997). Losartan and candesartan were made into generics at the very beginning of the 2010s. Until 2016, all significant ARBs were genericized. The innovative and generic market entries define market patterns in a very complex manner.

To describe the macro-level trends in the market performance, we used the methodology described above. Analyses of aggregate sales revenues of ARBs and ACE inhibitors were repeated on available data for 2001, the earliest year of the database, data for 2009, from the middle of the period, and 2016 as the end of a 15-year trend. The indices were created for comparison and multidimensional scaling was also carried out.

Considering the data for 2001 (Table 1), every market was dominated by ACE inhibitors. Two groups can be defined: the first group includes Hungary, Poland, and Romania, with practically no ARB sales; the second group involves Western countries, with ARB sales volumes ranging between 14 and 37%. In the first group of countries, ARBs were much more expensive than ACEIs (four to eight times more), while in the Western group, ARBs were also more expensive on average, but to a lesser extent, if we compare the relative prices. All ARB preference indices were below “1” in 2001 as opposed to in 2009, despite the fact that ARBs accounted for over 40% of the sales revenue in France and Spain.

Assessing the data from 2009 shows that ARB relative prices were conspicuously higher in the investigated group of countries (Table 2). For instance, the difference in cost between the two drug groups varied by a factor of ten in Germany and in the UK. Presumably as a consequence, the volume share of ARBs exceeded that of the ACE inhibitors only in two countries (France and Spain). The market share of ACE inhibitors accounted for 70–90% of the DDD-adjusted sales volume in Germany, Hungary, Poland, Romania, and the UK, and in the three Central Eastern European countries this drug group accounted for the majority of manufacturing revenues, too. With the exception of the UK, the preference index exceeded “1” in every Western European country, with Spain, the Netherlands, and France having the highest value. ARBs were more expensive in 2009 than in 2016, and in most Western European countries with similar volume shares, shares of manufacturing revenue were also higher.

In general terms—based on DDD-adjusted sales figures in 2016—the volume share of ACE inhibitors was greater than that of ARBs, so the trend for increasing ARB sales that we observed between 2001 and 2009 did not continue (Table 3). France, Spain, the Netherlands and Italy accounted for the greatest ARB share of DDD-adjusted volume, exceeding 50% in the first two countries, while the market share in Italy and the Netherlands was more than 40%. In contrast, ACE inhibitors accounted for more than 70% of the DDD-adjusted sales volume in Poland, UK, and Romania. In Poland and Romania, ACE inhibitors also accounted for the majority of manufacturing revenue, similarly to in Hungary, the third Central and Eastern European country. In other countries, ARBs accounted for the larger share of manufacturing revenue. In Spain and Germany, ARBs accounted for over three-quarters of the sales revenue of the whole drug group. The difference in average price level of ACE inhibitors and ARBs—a possible cause and effect of these effects—was also highest in these two countries, as was the ARB preference index. The ARB preference index, reflecting the relative sales and estimated price levels of the two subgroups together, was 1.95 and 5.83 in the German and Spanish market, respectively, indicating significant ARB sales despite their high price in relation to that of ACEIs. A value of around “1” for France, Italy, and the Netherlands indicates balanced market conditions, while a value in the range of 0.1–0.3 for Poland, Hungary, and the UK indicates the market dominance of ACE inhibitors. ACE inhibitors were generally cheaper than ARBs in all nine of the countries, although the price ratio shows that the gap clearly closed between 2009 and 2016.

Comparison of the results of multidimensional scaling for 2016, 2009, and 2001 (Figure 1) shows the clear dominance of ACEI therapies in 2001 (countries are close to “0” in relation to dimension 2), with ARBs having been available for only a few years. Until 2009, a trend for the increasing use of ARBs can be seen, which may be explained by their acceptance, and, by 2009, the maturity of innovative ARB brands. The increase in ARB sales happened in spite of the significant price index increase between ACEIs and ARBs (with ARBs being 1.83–10.64 times more expensive than ACEIs in different countries). The reason for this is the genericization of ACEI brands, which pushed prices down, while ARB active ingredients maintained their innovative status and monopoly. Accordingly, competition in the ACEI markets occurred at the level of brands, and in ARB markets at the level of APIs. Until 2016, both ARB and ACE prices dropped significantly, and the scissor of the price index between the two groups also closed to a range of 1.13–5.11, still in favour of ARBs. Interestingly, although the price level of ARBs decreased and their relative price compared to ACEIs also decreased in the examined period, the dominance of ARB sales volumes did not follow this trend, ending up with almost the same volume ratio between ACEIs and ARBs in 2016 as in 2009. For this reason, the multidimensional scaling indicates a closing pattern.

Considering the countries in the study, it is clear that although the relative position of Western European countries changed, the situation of the three Eastern European countries—in which the lower level of use of ARBs is reflected—remained relatively similar and differentiated from the Western countries. However, in 2009, before the launch of generic ARBs, the German and the British markets formed one group identifiable by less frequent use of ARBs. Additionally, the Italian and French as well as the Dutch and Spanish country pairs can also be distinguished in 2009. These pairs, comparing the figures from the tables above, had very similar relative prices for ARBs and the ACEI group, and the pairs also had a very similar share of the ARB market. The stress index fits the data for all years well.

The trends can be better assessed, and countries can be more sharply differentiated if the ARB preference index and price levels are plotted over time, year by year (Figure 2). In the UK, Hungary, Poland, and Romania, the preference index remained invariably low in the period under analysis. In contrast, despite the high prices, ARBs were responsible for a significant share of the market in the other five countries, leading to a peak in the preference indices curve in around 2010, before the drop in ARB prices.

Along with the drastic price decline in generic drugs, the market share of ARBs increased in the three Central Eastern European countries and Germany; however, in France, Spain, Italy, and the Netherlands, in around 2010, when prices started to drop, the volume share of ARBs stopped growing. It should be emphasized that the average price level of ARBs in all countries was higher than that of ACE inhibitors during the whole period, but the relative price difference between ARBs and ACE inhibitors tended to decrease in all countries. In Hungary and the Netherlands, the price of ARBs has been close to that of ACE inhibitors in recent years, and in Romania and Poland, price levels have also become significantly closer. The biggest gap between price levels remains in the Spanish market.

### 3.2. Preferences for ACE Inhibitor Active Ingredients in Six European Countries

After the comparison of the two therapeutic groups, the evaluation now changes the focus to ACEI therapies. Kovács and Simon assumed a positive correlation between clinical evidence and market performance, presumably leading to ramipril sales outperforming other drugs in terms of volume. To better understand the trends in market share changes and the reasons behind them, we analyse the market factors of ACEI therapies in the period under review. In addition to the three Central Eastern European countries, we compare performance trends in Germany, France, and the UK. In the examination of active ingredients, we help to interpret the diffusion patterns by assessing the number of brands, volume shares, and price levels.

The sales volume of ACE inhibitors generally increased in the selected European countries in the examined period. The market shares for DDD-adjusted volumes (Figure 3) reveal that the diffusion of dominant drugs formed the market landscape in the investigated period. By 2016, with the exception of Hungary and Romania (where perindopril was the most popular drug), ramipril became the market-leading therapy. In the UK, ramipril has been responsible for the largest market share since the mid-2000s. However, in Germany and Poland, following 2007–2008, it replaced the former market-leading drug enalapril within a few years. In France, the sales volume of perindopril approached that of ramipril, and since about 2008, the market share of the two active ingredients has stabilized. Similarly, the market share of perindopril has also been increasing in Hungary and Romania since 2008. The market share of enalapril was very significant in Poland, Hungary, Germany, and Romania in the early 2000s, but afterwards decreased significantly in almost all countries. Captopril, the first ACEI, also had a large market share in the Romanian and German market at the dawn of the 2000s, but declined in significance by the end of the period. Lisinopril had a larger market share primarily in France, Germany, and the UK, but with the exception of the UK, its relative market share decreased. It is surprising that, in contrast to in other countries, the relative sales of various active ingredients have not changed much in France and Poland since 2011–2012, apart from the slow decrease in sales of lisinopril and enalapril in favour of perindopril and ramipril. Interestingly, the growth rate of the market share of ramipril slowed down temporarily for a period of three or four years after 2004, when the share of perindopril increased.

Apart from the above-discussed products, other active ingredients assumed minor significance only, yet some country-specific features are worth mentioning. Trandolapril had a noticeable market share in the French and Romanian market in the early 2000s, and zofenopril only in Romania, with a very small market share in France and Poland. Cilazapril had a noticeable market share mainly in the Polish market, quinapril in the Polish, Romanian, and French markets, and fosinopril in the Hungarian, Romanian, and French markets.

The market performance of active ingredients is defined by multiple factors as we described it in in the literature review. To establish a clear picture, price levels, number of brands, and DDD-adjusted volume shares of the five most significant drug therapies in terms of market share—captopril, enalapril, lisinopril, perindopril, and ramipril—are plotted in Figure 4. This graph shows that the prices of the most frequently used ACE inhibitors decreased between 2001 and 2016. The inversely proportional impact of the increase in the number of brands on price levels can also be confirmed, with ramipril falling significantly in every market, but perindopril falling less significantly relative to ramipril, with almost no change in Germany. The price of enalapril has not changed in France, nor zofenopril in France and Romania, nor lisinopril in Hungary. Considering prices in general, those for active ingredients with lower sales volumes tended to be higher. Ultimately, the price of ramipril was among the lowest in every market during the last few years of the period under analysis.

The total number of ACE inhibitor brands continued to increase until around 2010, before declining or stagnating in most countries. In Germany, the number of brands topped out somewhat earlier, in around 2007, but then decreased faster than in other countries due to the sharp decline in the number of captopril brands. The number of perindopril brands soared between 2008 and 2010 (except for in Germany), after which active ingredients with larger market shares (captopril, lisinopril, perindopril, ramipril, enalapril, trandolapril, quinapril, and fosinopril) all became members of a multi-player market. This coincided with the period when the number of ARB active ingredient brands started to increase. In contrast, the price of single-drug ACE inhibitors—with the exception of perindopril—started to decline in most countries in around 2005, with an increase in the number of brands concurrently, followed by combination drugs a few years later.

In most countries, the price and the market share of captopril, enalapril, and lisinopril continuously declined over the period, while the number of brands of enalapril and captopril started to grow in the early 2000s, then more or less stabilized and then started to decline. Unlike in other countries, the price of captopril in Hungary slowly increased until 2010, and in the UK, prices shot up between 2013 and 2015 (the paradox of generics). One feature of lisinopril should be emphasized: the Polish and the Romanian markets show a slight increase in the market share after the fall in prices in around 2005. In France, the fall in prices and sharp increase in the number of brands of perindopril and ramipril indicates the market entry of generic drugs in around 2005 and 2008. Before that, the market share of the two active ingredients increased, but remained essentially unchanged for the remainder of the period. In the German, Polish, and British markets, the volume share of perindopril did not change significantly, despite the fall in prices, but the market share of ramipril increased rapidly following the decrease in prices and the growth in brand numbers in the three markets mentioned above, and in Hungary and Romania as well. Interestingly, in the UK, the growth rate fell slower than previously. However, the market share of perindopril grew steadily in the Hungarian and Romanian markets. Here, despite the rapid rise in the number of brands, prices declined at a slower rate.

Market fragmentation or concentration is best measured by the Herfindahl–Hirschman index (HHI). Annual HHI scores for the markets of ACE inhibitor active ingredients in six countries are presented in Figure 5. The figure shows the outcomes of the DDD-adjusted volume- and revenue-based calculations; furthermore, reference raw calculations—that is, correction-free daily dose calculations—are also presented. With the strengthening market position of ramipril in Western countries, and that of perindopril in France, it is not surprising that the HHI scores continuously grew in these countries. Concurrently, relatively concentrated markets started to become increasingly fragmented and HHI declined in CEE countries by the end of the 2000s, before starting to rise again with the growing dominance of ramipril and perindopril. Interestingly, the HHI scores of raw and DDD-adjusted volume figures started to diverge increasingly in Germany, the UK, and Poland, where ramipril had the largest market share. The reason for this may be that ramipril, in contrast to perindopril, is used in much higher doses than DDD in clinical practice. As with the market share figures, market concentration was no longer on the rise in Poland after 2011–2012.

## 4. Discussion, Conclusions, Practical Implications

The diffusion of innovative and generic drugs and changes in prescription preferences involves a complex process in which market positions are defined by the pharmaceutical market players and the characteristics of competing drugs. One of the characteristics of the pharmaceutical market is that only similarly or equally effective innovative products can compete with each other until the expiry of their patents or market exclusivity. This phenomenon can be clearly seen if we consider the decline in the use of captopril, the first line of therapy of the two groups, but an insignificant player by the end of the examined period. Captopril was outperformed by better candidates over the decades.

In the generic market for active ingredients, decades-old innovations are present and access to them is improving due to falling prices and an increasing number of market players. This also means that there is much more knowledge about a few active ingredients (and whole groups of them) than about entirely new, pioneering therapies—so, according to Kovács and Simon, generic markets may play a major role in the spread of better therapies from a clinical point of view. Considering aggregate, long-term change in sales volumes, two conclusions can be drawn. A comparison of ARB and ACEI therapies shows that the increasing dominance of ARBs in European markets cannot be identified, which, in spite of the decreasing price and clinical superiority of ARBs, may be a negative outcome from the patient’s perspective. On the contrary, in terms of ACEI sales volumes, ramipril has been the most popular option, which is a positive outcome from the same perspective based on a ranking of evidence-based principles.

One of the most striking features of the time-series data is that in Hungary, Romania, and Poland, the sales revenues of the ARB drug group increased in line with the emergence of generic drugs—even if slowly and from a low baseline. However, and except for Germany, this is not the case for the Western European countries. In the case of the latter, with the diffusion of generic drugs, ARB sales stagnated in the 2010s, and the market share in relation to ACE inhibitors remained virtually unchanged, despite the decline in ARB prices. It seems that the initial spread of ARBs was not hindered by high prices in most Western countries, but the availability of generic drugs improved access to ARBs in the German and three Eastern European markets. An explanation for the flattening volume curves of ARBs and ACE inhibitors in Western countries is that Western markets that started to use ARBs earlier became saturated, meaning that the number of untreated patients requiring angiotensin medication had become very small by the end of the period. This may be true for the Netherlands, the UK, and Italy, where the sales data of both ACE inhibitors and ARBs, after reaching a plateau, remained largely unchanged for a longer period. In contrast, although ARB sales stagnated in Spain and France after 2010, sales of ACE inhibitors increased almost linearly. An important factor behind this trend is hypothesized to be the difference between the level of competition between the two drug groups. Until the 2010s, ARBs, associated with a monopoly and high prices, competed on the level of the API. In contrast, ACE therapies started to compete on the brand level with multiple generic entries, pushing prices down and boosting diffusion in relation to ARB therapies. Accordingly, the diffusion of the therapeutic groups seems to have been strongly affected by the difference between the level of competition. Price sensitivity, drug policies, and regulations are thought to be key additional factors that define diffusion at the country level.

In the 2000s, increasing expenditure both in France and Spain led to regulations aimed at reducing spending on pharmaceuticals [55,56]. Looking at ARB volume shares in 2016 again, it can be seen that they were the highest in France and Spain: 50.63 and 51.66%, respectively (slightly lower than in 2009). In contrast, the UK had the lowest share in 2016: 22.71%. In the UK market, according to the NICE recommendation [57,58], ARB should only have been prescribed in the case of intolerance or contraindication to ACE inhibitors before 2011, but the 2011 directive [59] also recommended the use of cheap ARBs as a first-line approach. Nevertheless, the market share of ARBs in the UK remained essentially unchanged after 2005. The almost unchanged mid-term market share suggests that prescribing habits are fixed, at least at the therapeutic group level. One limitation of this study is that the role of promotional activities cannot be explained from the data, and thus should be further investigated.

Turning the focus to the investigation of ACEI therapies, the most significant active ingredients became generic ones in the nine countries in the 2000s. Competition existed mainly among generic drugs, even if active ingredients with low sales had no or very few competitors on European markets. Thus, data for the first half of 2016 reflect well-developed generic market conditions, if not necessarily perfect competition. Based on our results, we draw the conclusion that ramipril outperformed its competitors in the long term. By 2016, with the exception of Hungary and Romania, where perindopril was the most popular drug, ramipril continuously became the market-leading therapy. The European markets that were investigated by the end of the period can be symbolized by the competition between ramipril and perindopril, with two other compounds also competing for sales (lisinopril and enalapril), and others declining in significance. The diffusion of clinically preferred therapies was strongly supported by the generic entries, and the inversely proportional impact of the increase in the number of brands on the price level can also be confirmed: the price of ramipril fell significantly in every market, but that of perindopril fell less significantly relative to ramipril, with almost no change in Germany.

Interpreting the results with Denig’s model, in countries such as France, Spain, Italy, and the Netherlands, it is more likely that drugs enter the typical medical ‘evoked set’ start before genericization, while in other countries such as Poland, Romania, and Hungary, this process occurs after genericization. Alternatively, and as an analogy of Rogers’ model of innovation diffusion, Eastern European countries may be perceived as late adopters of new ACE inhibitors and ARBs. However, it is doubtful as to whether the existence of late users is really related to uncertainty about new technology, as suggested by Rogers, not to the pricing of medicines. The phenomenon that manufacturers tend to enter markets with higher potential should also be considered—this defines the order of penetration, starting from Western countries towards Eastern ones. The significance of this phenomenon is nowadays outweighed by regulatory issues, since new chemical entities must be authorized through a centralized procedure, but this was not the case when the compounds we investigated were registered.

The impact of marketing activities can only be assumed with limitations due to the aggregate data. Country-origin effects suggested by Kovács and Simon cannot be excluded for perindopril or lisinopril, even if long-term sales revenues are examined. The market share of perindopril in the French, Romanian, and Hungarian markets was significant for most of the period examined. On the other hand, the rate of market share change was slightly different: in France, the share remained unchanged from 2009 onwards, while in the Hungarian market, the share of perindopril accelerated from around the same time, while in Romania growth has been almost linear since the early 2000s. The number of perindopril brands on the market increased dramatically between 2008 and 2010, suggesting that the emergence of generic competition was more conductive to the diffusion of active ingredients in Hungary, but in France, similarly to ramipril, growth was interrupted following the emergence of perindopril generic drugs. In Romania, the entry of generic drugs had little effect on the market dynamics of perindopril diffusion. At the same time, in the Hungarian, Polish, and Romanian markets, the market share of ramipril, similarly to that of ARBs, increased along with price reductions and the emergence of generic competition. This suggests that in the three Eastern European countries, the proliferation of new therapies was less influenced by manufacturers’ promotions than by price competition as a result of the entry of generic drugs, allowing for the wider use of more advanced treatments. Interestingly, even in the cost-conscious UK market that was dominated strongly by ramipril, revenue-based data suggest that, until genericization, the promotion of perindopril slowed down the diffusion of the generic and cheap ramipril. When assessing the competition, we should not forget the lawsuit related to perindopril. The violation of competition, alongside the other important factors described above, may have had an important impact on market performance that we can observe at the end of the period.

In Eastern European countries, the emergence of generic markets has not only improved access to ACE inhibitors and ARBs, but has been a prerequisite for changing preferences. In contrast, genericization resulted in the relative anchoring of preferences in some Western European countries, which phenomenon may be attributed to the role of the cessation of promotion.

The temporal change in the Herfindahl–Hirschman index is somewhat similar in the three CEE countries, especially when compared to that for the Western countries. From the early 2000s onwards, the HHI for both manufacturing revenues and sales revenues indicates a declining concentration, which reverses after 2006–2007. In contrast, French, German, and British data show a near-constant increase in market concentration. With regard to the market share of active ingredients, this is primarily due to the steady decline in the large share of enalapril (and captopril in Romania) in the early 2000s. Thus, the market share of newer therapies (ramipril, lisinopril, and perindopril) was greater in the three Western countries compared to in Eastern European countries, where initially the older and cheaper active ingredients were more prevalent.

A limitation of the study is that, although goods can move freely within the EU, healthcare financing is essentially within the jurisdiction of member states; furthermore, Hungary, Poland, and Romania joined the EU during the period under examination (the effect of parallel imports was also not taken into account). In addition, many other political and economic changes have taken place in the countries during this period, and health reforms and the economic crisis of 2007–2008 limit the interpretability of the data presented here. Additionally, survey data do not cover the entire European Union, which also imposes limits on the generalizability of conclusions. Finally, it should be mentioned again that long-term aggregate sales data were analysed, without taking pharmaceutical promotion or differences in approved indications into deeper consideration, and so only assumptions can be made about the latter relationships. We admit that the correlation between the clinical evidence and the market performance is strongly biased by multiple factors where the prescribing physician plays the key role. The assessment of the different factors that affect the perception of the physician is the key to understanding the trends that we describe in the current study. Based on the results, the authors suggest further in-depth investigation of the multi-factorial relationships that finally leads to drug prescriptions.

In terms of practical impacts, it is recommended that innovative pharmaceutical companies, when designing their promotional activities, assess and consider market characteristics and limit their spending in markets where the entry of generics is more permissive. However, in markets where several ‘me-too’ players are concurrently present, generic manufacturers should pay attention to the promotion of innovators, in addition to the sales trends of the last few years prior to the termination of market exclusivity, since when the promotion of the entry of generic drugs is cut back, revenues from sales of active ingredients may not increase as expected. Additionally, it may be worthwhile to conduct research into evoked sets among potential prescribers to map out their prescription patterns. Considering the case of ramipril, generic manufacturers should seek to identify drug therapies with better safety and efficacy profiles that, in the long term, can contribute to more sales and higher profits.

Furthermore, it may be beneficial for funding institutions, along with medical and pharmaceutical professional organizations, to formulate national recommendations even within drug classes at an active substance level to reduce expenditure on and promote the uptake of more advanced medicines in generic markets.

## Figures and Tables

**Figure 1 ijerph-18-09429-f001:**
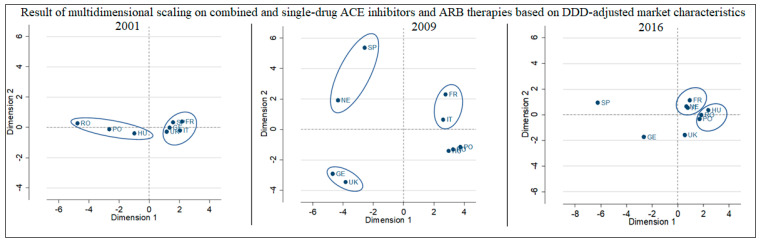
Result of multidimensional scaling of combined and single-drug ACE inhibitors and ARBs based on DDD-adjusted market characteristics of 2001, 2009 and 2016: 2001: r = 1.0000, ρ = 0.9984, Kruskal’s stress index: 0.0056; 2009: r = 1.0000, ρ = 0.9999, Kruskal’s stress index: 0.0045; 2016: r = 1.0000, ρ = 0.9990, Kruskal’s stress index: 0.0048.

**Figure 2 ijerph-18-09429-f002:**
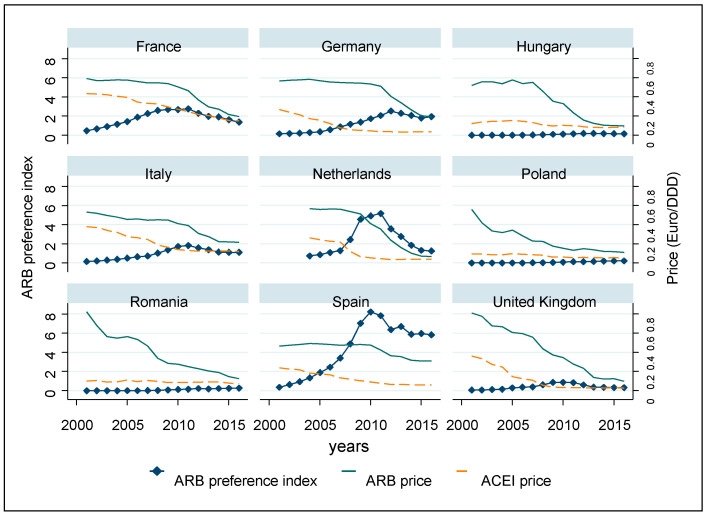
ARB preference index and average ARB and ACEI prices.

**Figure 3 ijerph-18-09429-f003:**
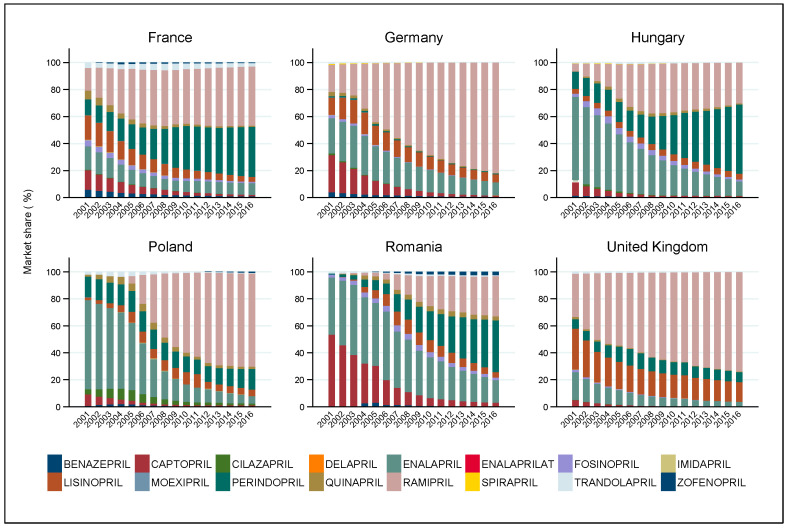
Volume share of ACE inhibitors (combination drugs and single-drug preparations) in six European countries between 2001 and 2016.

**Figure 4 ijerph-18-09429-f004:**
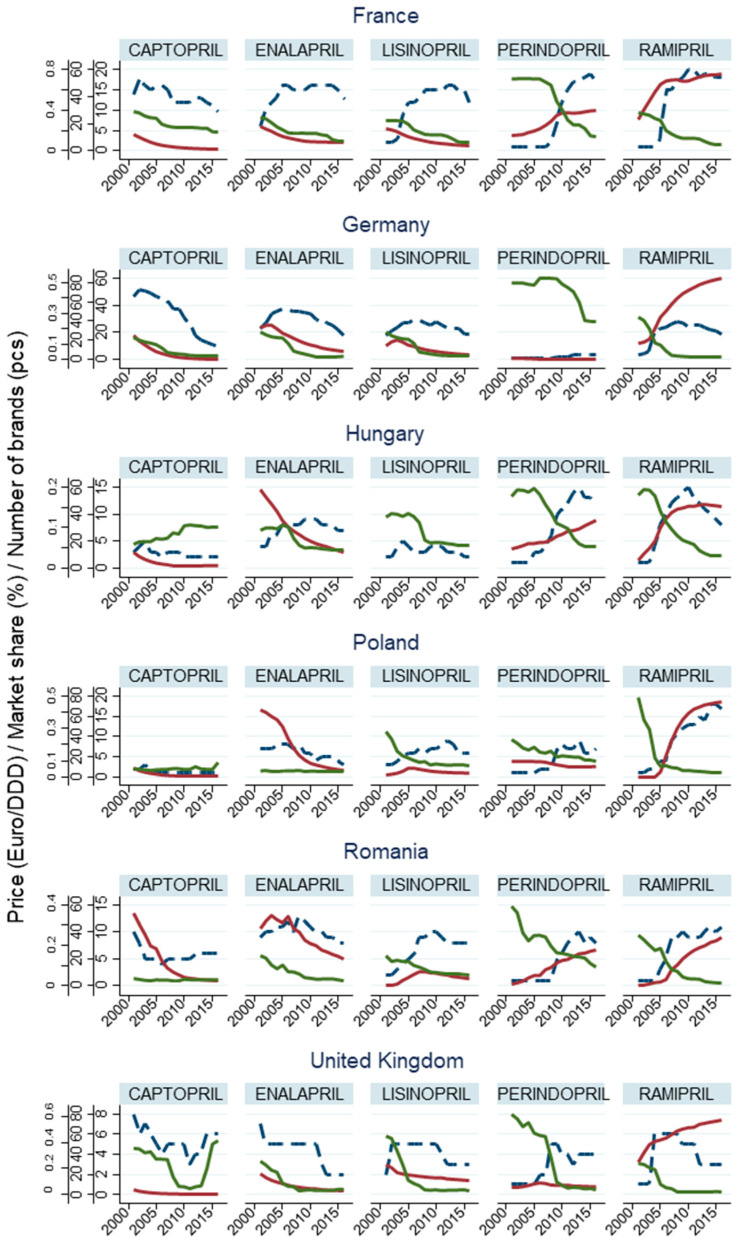
Changes in sales, price, and number of brands of single-drug ACE inhibitors (including the five with the most significant market share). Blue broken line (- - -): number of brands, red straight line (-): DDD-adjusted volume share, green straight line (-): price of defined daily dose.

**Figure 5 ijerph-18-09429-f005:**
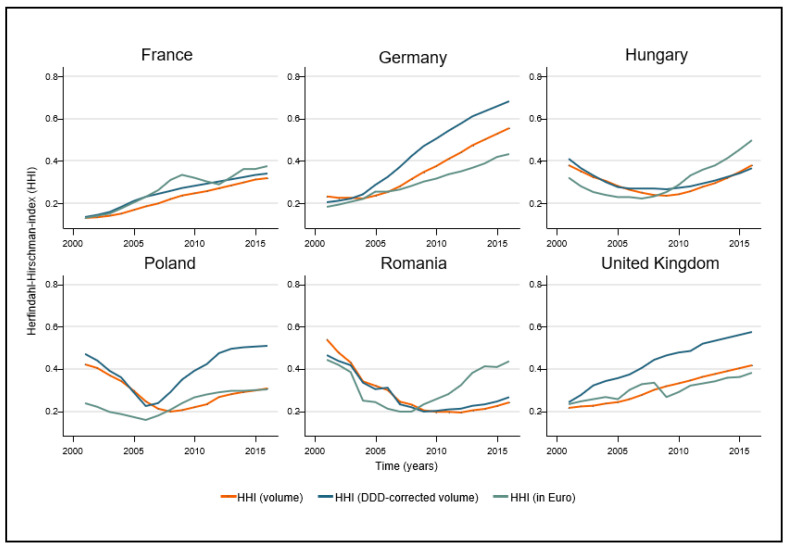
Market concentration of ACE inhibitor active ingredients in six European countries from 2001 to 2016.

**Table 1 ijerph-18-09429-t001:** Market features of ARBs and ACE inhibitors in 2001 based on DDD-adjusted volumes, taking into consideration combination drugs and single-drug preparations.

	ARB Price Level (EUR/DDD)	ACEI Price Level (EUR/DDD)	ARB Volume (DDD. %)	ARB Sales Revenue (EUR. %)	ACEI Volume (DDD. %)	ACEI Sales Revenue (EUR. %)	ARB: ACEI Price Level Ratio	ARB: ACEI Volume Ratio	ARB: ACEI Revenue Ratio	ARB Preference Index
FR	0.59	0.43	37.34	44.85	62.66	55.15	1.36	0.60	0.81	0.48
GE	0.57	0.27	21.61	36.81	78.39	63.19	2.11	0.28	0.58	0.16
HU	0.52	0.12	1.03	4.31	98.97	95.69	4.32	0.01	0.05	0.00
IT	0.53	0.38	24.75	31.56	75.25	68.44	1.40	0.33	0.46	0.15
PO	0.56	0.09	0.19	1.16	99.81	98.84	6.01	0.00	0.01	0.00
RO	0.82	0.10	0.09	0.69	99.91	99.31	8.16	0.00	0.01	0.00
SP	0.46	0.24	29.99	45.47	70.01	54.53	1.95	0.43	0.83	0.36
UK	0.81	0.36	14.23	27.11	85.77	72.89	2.24	0.17	0.37	0.06

Abbreviations: FR—France, GE—Germany, HU—Hungary, IT—Italy, PO—Poland, RO—Romania, SP—Spain, UK—United Kingdom.

**Table 2 ijerph-18-09429-t002:** Market features of ARBs and ACE inhibitors in 2009 based on DDD-adjusted volumes, taking into consideration combination drugs and single-drug preparations.

	ARB Price Level (EUR/DDD)	ACEI Price Level (EUR/DDD)	ARB Volume (DDD. %)	ARB Sales Revenue (EUR. %)	ACEI Volume (DDD. %)	ACEI Sales Revenue (EUR. %)	ARB: ACEI Price Level Ratio	ARB: ACEI Volume Ratio	ARB: ACEI Revenue Ratio	ARB Preference Index
FR	0.54	0.29	54.79	68.92	45.21	31.08	1.83	1.21	2.22	2.69
GE	0.54	0.05	26.35	79.20	73.65	20.80	10.64	0.36	3.81	1.36
HU	0.36	0.10	14.32	37.93	85.68	62.07	3.66	0.17	0.61	0.10
IT	0.45	0.16	41.44	65.94	58.56	34.06	2.74	0.71	1.94	1.37
NE	0.51	0.07	43.55	85.52	56.45	14.48	7.66	0.77	5.91	4.56
PO	0.17	0.06	11.30	26.86	88.70	73.14	2.88	0.13	0.37	0.05
RO	0.28	0.08	13.35	34.06	86.65	65.94	3.35	0.15	0.52	0.08
SP	0.48	0.10	55.30	85.05	44.70	14.95	4.60	1.24	5.69	7.04
UK	0.37	0.04	22.32	74.76	77.68	25.24	10.31	0.29	2.96	0.85

Abbreviations: FR—France, GE—Germany, HU—Hungary, IT—Italy, NE—The Netherlands, PO—Poland, RO—Romania, SP—Spain, UK—United Kingdom.

**Table 3 ijerph-18-09429-t003:** Market features of ARBs and ACE inhibitors in 2016 considering adjusted volume of combination drugs and single-drug preparations.

	ARB Price Level (EUR/DDD)	ACEI Price Level (EUR/DDD)	ARB Volume (DDD, %)	ARB Sales Revenue (EUR, %)	ACEI Volume (DDD, %)	ACEI Sales Revenue (EUR, %)	ARB: ACEI Price Level Ratio	ARB: ACEI Volume Ratio	ARB: ACEI Sales Revenue Ratio	ARB Preference Index
FR	0.20	0.15	50.63	57.04	49.37	42.96	1.29	1.03	1.33	1.36
GE	0.19	0.04	37.56	76.44	62.44	23.56	5.39	0.60	3.24	1.95
HU	0.10	0.09	27.07	29.48	72.93	70.52	1.13	0.37	0.42	0.16
IT	0.22	0.12	43.85	58.65	56.15	41.35	1.82	0.78	1.42	1.11
NE	0.07	0.04	45.42	59.88	54.58	40.12	1.79	0.83	1.49	1.24
PO	0.11	0.05	25.04	40.69	74.96	59.31	2.05	0.33	0.69	0.23
RO	0.12	0.07	28.44	40.61	71.56	59.39	1.72	0.40	0.68	0.27
SP	0.31	0.06	51.66	84.52	48.34	15.48	5.11	1.07	5.46	5.83
UK	0.10	0.03	22.71	52.12	77.29	47.88	3.70	0.29	1.09	0.32

Abbreviations: FR—France, GE—Germany, HU—Hungary, IT—Italy, NE—The Netherlands, PO—Poland, RO—Romania, SP—Spain, UK—United Kingdom.

## Data Availability

Restrictions apply to the availability of the data presented in the article. Data was obtained from the MIDAS database and are available with the permission of IQVIA.

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
