# Peer review of "Drugs Becoming Generics—The Impact of Genericization on the Market Performance of Antihypertensive Active Pharmaceutical Ingredients"

_ijerph, 2021, doi:10.3390/ijerph18189429_

Round 1
Reviewer 1 Report
The manuscript is greatly improved and is recommended for publication.
Reviewer 2 Report
It is much more improved. I would only suggest to present the currencies in US Dollars as you can have potential readers from other regions and countries. In the sup´plementary material you present data (GDP per caíta) in US dollars (page 4) and in page 5-6 you present costs in Euros.
This manuscript is a resubmission of an earlier submission. The following is a list of the peer review reports and author responses from that submission.
Round 1
Reviewer 1 Report
In page 8 and in table 1 you mention only eight countries. Include in all tables the countries´ abbreviations. I would like the authors to give additional reasons for the ARB preference index and average ARB and ACEI price sensitivity, and the rest of the figures 3-5, which could influence that prices. For example, how was the the rate of heart attacks in the 9 countries during the period of study 2001-2016 and if that could affect such prices and the market share? or another secondary information regarding the illnesses and the use of these drugs. The authors forgot to mention why they choose these drugs more than just the pharmaceutical reasons, if it was because it attends the top illnes in the EU region? As the authors stated, prescribing drugs is multifactorial phenomena, so besides the ARB and ACEI priceS, they should include another macroeconomic variables such as the income level, inflation rates or GDP in the countries selected, and how that could affect such prices; or, how can you isolated the drugs prices from those variables in such period?? That might explain the differences between Western and Eastern EU (I would say their income level takes Eastern to be latter adopter), and the decline/ increase of prices; how could those macrovariables affect each drug´s price in each country and region? , so "the market position is not only defined by the pharmaceutical players and the characteristics of the competing durgs". Additionally, I consider important to justify why did you selected those 9 countries? was it because their cultural similarities, income level, or rates of specific illnesses? You should include that in a paragraph. In the discussion I would like to see the connection between prescribing the antihypertensive drugs by physicians that appears in the introduction.. Is the selection of drugs done taking exclusively the countries´ prices you show in figure 2- 5 , or how can the physicians could affect these prices (or is just the opposite?)
Reviewer 2 Report
Please add ACEI AND ARBs in the title.
Abbreviate ACE/ARB when appearing for the time.
The research involved? Or involves? Check grammar and syntax. Sentences are not clear.
Can the authors please describe the methods clearer in the abstract?
The abstract is not clear and the authors did not define and described the type of competition. For example, what is API or brand? Are they comparing to?
Do authors had a comparator such as standard drug (Patented drug?).
If authors are confused with several terms such as ramipril (what is the brand or generic competitor they compare to?)
Although the topic is interesting, it lack clarity and consistency.
- How impact was assessed (In general impact refers to intervention, I did not find any intervention in the study). Thus, the statements are confusing.
- The authors need to write the sentences clearly. The present format of the manuscript is uncertain.
- It was not clear how the long-term trend was assessed and how the authors defined the performance.
- How many are considered from each European region was not clear.
Introduction: Avoid using personal statements and descriptions. Frame the background based on evidence and describe the changing trends in antihypertensive medication, how they affect the outcomes, bioequivalence, and cost, etc.
The introduction section is too lengthy and It should be trimmed to 500-600 words specific to the study objectives.
Methods: The methodology section is not clear. The authors presenting a recent study that did not provide any reference and how these 9 countries were selected was not clear. The authors mentioned nine European countries in the abstract and Included the USA in the methods. Which is not an EU country.
The methodology section should be better organized. Overall, the authors confused with several personal statements and references with databases that did not provided how sales data was extracted or retrieved. How the data was validated.
Confusing definitions: “The daily defined dose (DDD) was defined on the basis of DDD data published by WHO and DDD correction was made to drug brands with various mechanism”. The authors did not define or described what is DDD means.
The manuscript has several syntax errors and long sentences. The manuscript requires English language editing.
Overall, I did not recommend the manuscript for publication as it is not clear and not suitable for publication as it is not within the scope of IJERPH.